# Exposition of Intermediate Hosts of Schistosomes to Niclosamide (Bayluscide WP 70) Revealed Significant Variations in Mortality Rates: Implications for Vector Control

**DOI:** 10.3390/ijerph191912873

**Published:** 2022-10-08

**Authors:** Alvine Christelle Kengne Fokam, Laurentine Sumo, Mohamed Bagayan, Hugues Clotaire Nana-Djeunga, Thomas Kuete, Gabriella S. Ondoua Nganjou, Murielle Carole Tchami Mbagnia, Linda Djune-Yemeli, Charles Sinclair Wondji, Flobert Njiokou

**Affiliations:** 1Department of Animal Biology and Physiology, Faculty of Science, University of Yaoundé 1, Yaoundé P.O. Box 812, Cameroon; 2Centre for Research in Infectious Diseases (CRID), Yaoundé P.O. Box 13591, Cameroon; 3Department of Biological Sciences, Faculty of Science, University of Bamenda, Bambili P.O. Box 39, Cameroon; 4Animal Biology and Ecology Laboratory, University of Joseph Ki-Zerbo, Ouagadougou P.O. Box 7021, Burkina Faso; 5Centre for Research on Filariasis and Other Tropical Diseases (CRFilMT), Yaoundé P.O. Box 5797, Cameroon; 6Department of Biological Sciences, Faculty of Medicine and Pharmaceutical Sciences, University of Douala, Douala P.O. Box 24157, Cameroon; 7Department of Vector Biology, Liverpool School of Tropical Medicine, Pembroke Place, Liverpool L35QA, UK

**Keywords:** schistosomiasis, snail control, Niclosamide, lethal concentrations, Cameroon

## Abstract

(1) Background: Schistosomiasis remains a public health issue in Cameroon. Snail control using Niclosamide can prevent schistosome transmission. It is safe to determine lethal concentrations for the population. This study aimed at assessing the toxicity of Niclosamide on different developmental stages of snail populations; (2) Methods: Snails were collected, identified, and reared in the laboratory. Egg masses and adult snails were exposed to Niclosamide, at increasing concentrations (0.06, 0.125, 0.25, 0.5, 1 mg/L for egg embryos and 0.06, 0.08, 0.1, 0.12, 0.14, 0.16, 0.18, 0.2 mg/L for adults). After 24 h exposure, egg masses and snails were removed from Niclosamide solutions, washed with source water and observed; (3) Results: Snail susceptibility was species and population dependent. For egg embryos, *Biomphalaria pfeifferi* was the most susceptible (LC_50_: 0.1; LC_95_: 6.3 mg/L) and *Bulinus truncatus* the least susceptible (LC_50_: 4.035; LC_95_: 228.118 mg/L). However, for adults, *B. truncatus* was the most susceptible (mortality rate: 100%). The LC_50_ and LC_95_ for *Bi. camerunensis* eggs were 0.171 mg/L and 1.102 mg/L, respectively, and were higher than those obtained for adults (0.0357 mg/L and 0.9634 mg/L); (4) Conclusion: These findings will guide the design of vector control strategies targeting these snail species in Cameroon.

## 1. Introduction

Schistosomiasis is an acute and chronic parasitic disease, which constitutes a public health problem in poor and rural communities. Indeed, this disease is responsible for growth retardation and a delay in learning among children, as well as decreased ability to work among adults, thus representing an impediment to development. Schistosomiasis is endemic in 78 countries, and about 24,000 deaths and 2.5 million of DALY (disability-adjusted life years) have been reported in 2016. In 2018, the total number of people in need of preventive chemotherapy was 229.2 million, of which 124.4 million were school-aged children [1]. The major WHO strategy to eliminate schistosomiasis as a public health problem focuses on periodic and targeted treatment with praziquantel of affected populations, especially at-risk groups [1]. This repeated mass treatment campaigns have led to a significant decrease in infection prevalence in many countries.

In Cameroon, surveys conducted in 2010–2012 revealed hotspots for transmission of schistosomiasis, despite the high impact of preventive chemotherapy on prevalence and intensities of infection [2,3]. This situation prompted the Ministry of Public Health, with the support of its partners, to broaden chemotherapy to all age groups and implement and/or reinforce water, sanitation and hygiene (WASH) measures [4]. These intensified control measures contributed to further reduce the prevalence of the infection, though the transmission of the disease is still ongoing [5] (Kengne et al., unpublished data). Indeed, the pattern of reinfestation of villagers is quite high because of their extensive reliance on the infected water for daily activities and the presence of parasite reservoirs in human and snail populations, as infected humans are not systematically diagnosed and treated in endemic localities [6]. Additionally, infected snails are able to release the parasite continuously during one year in water bodies where humans attend to their daily activities [7,8]. This epidemiological situation calls for the implementation of alternative approaches to tackle the transmission of schistosomiasis and accelerate its elimination as a public health problem.

Control of intermediate snail hosts from local habitats may be efficient for the prevention of *Schistosoma* infection and/or reinfection [9,10]. In Cameroon, human schistosomiasis is focally distributed and transmitted by seven species of snail hosts widely distributed throughout the country [11], *Bulinus* and *Biomphalaria* intermediate host species being the most represented. *Bulinus truncatus* (Audouin, 1827), *Bulinus globosus* (Morelet, 1866), *Bulinus senegalensis* Muller, 1781 and *Bulinus camerunensis* Mandahl-Barth, 1957 transmit *Schistosoma haematobium* (Bilharz, 1852) (urinary schistosomiasis), whereas *Biomphalaria pfeifferi* (Krauss, 1848) and *Biomphalaria camerunensis* (Boettger, 1941) are intermediate hosts of *Schistosoma mansoni* Sambon, 1907 (intestinal schistosomiasis) and *Bulinus forskalii* (Ehrenberg, 1831), the intermediate host of *Schistosoma guineensis* (Pages, 2003) responsible of the rectal schistosomiasis. One approach to fight these hosts includes the use of Niclosamide (Bayluscide WP 70), which until now has been the mollusciciding strategy recommended by WHO [12]. Indeed, Niclosamide is quite efficient against snails and their egg embryos, and also against all parasite stages, including both schistosome eggs and free schistosome larvae found in infected water [11]. Specific Niclosamide concentration should be previously defined for a given population since variability of snail susceptibility to Niclosamide has been demonstrated [13].

It, therefore, appears necessary to assess the toxicity of Niclosamide on different developmental stages (egg embryos and adults) of snail populations, in order to determine the lethal concentrations before implementing snail control. Here, we aimed to experimentally test (i) specific lethal concentrations of Niclosamide on egg embryos from three snail species (ii) specific lethal concentrations of Niclosamide on adult snails from three species and (iii) investigate the avoidance behaviour of adult snails exposed to Niclosamide. This experimental study will provide valuable data necessary for a safe implementation of targeted snail control in Cameroon.

## 2. Materials and Methods

### 2.1. Study System

Snails were harvested from sites currently described as active foci of Schistosomiasis and/or sites where snails were previously collected and identified using morphological and molecular tools. Once in the molluscarium, snails were identified and natural infection was checked by keeping snails under an artificial light. Snails emitting trematode cercariae were removed from the studied samples. Egg embryos were collected from the uninfected wild snails in each population to (i) test their susceptibility to Niclosamide (ii) to develop the G1 snail generation on which susceptibility of adults snails was tested. The number of egg masses and adult snails tested was identified by convenience, based on WHO guidelines for laboratory and field testing of molluscicides for snail control in schistosomiasis [12].

### 2.2. Collection and Rearing of Snails

Malacological surveys were carried out on February 2021 in Makenene Mock River (04°53.123′ N 10°47.206′ E) (Centre Region); in Petpenoun male and female lakes (05°38.087′ N; 10°38.139′ E), Monoun Njindoum Lake (05°34.895′ N; 10°35.382′ E) and Mangoun River (05°29.955′ N; 10°36.504′ E) (West Region). Makenene Mock River is a currently active focus of schistosomiasis where transmission is ongoing despite many years of chemotherapy [4,5] (Kengne-Fokam et al., unpublished data), where a target snail control is thought to be applied. Petpenoun, Monoun Njindoum Lakes and Mangoun River are previously visited snails habitats where molecular tools were used to identify Biomphalaria snail species [8,14]; Bulinus snail species of these sites have been identified using shell morphology. Snails were sampled by two collectors per hour, using a 2 mm-latticed steel sieve mounted on a 1.5 m wooden handle used to comb aquatic vegetation, or by hand picking from the mud with soft forceps [8]. Collected snails were identified using shell morphological keys defined by Chappell [15] and transported to a molluscarium at the Faculty of Science of the University of Yaoundé 1. Depending on their species and sampling sites, snails were organised into six populations: (i) *Biomphalaria pfeifferi* from Mock River, (ii) *Biomphalaria camerunensis* from Petpenoun Female Lake and Mangoun River and (iii) *Bulinus truncatus* from Petpenoun Male Lake, Monoun Njindoum Lake and Mangoun River. The room was thermoregulated with a portable air conditioner at 26 ± 1 °C and snails were maintained under a 12 L/12 D photoperiod throughout the experiment.

Wild snails (G0) from each population were kept together in 1.5 L plastic boxes containing water for acclimatization. A natural source water, which emerged from the rocks in the neighbouring area of the university, was used for rearing. Once in the laboratory, this water was filtered with a 12 L bench top water purifier system. After being screened for cercarial production, mature individuals from each population and each species were randomly chosen and isolated per group of three individuals in at least 15 waterproof plastic boxes of 300 mL for rearing purpose. Small white pieces of polystyrene were introduced in the rearing boxes for egg capsules laying [16]. Egg masses were then collected from each of the plastic boxes for susceptibility assay or for the production of adults (G1). During the rearing process, snails were fed ad libitum with fresh lettuce (*Lactuca sativa*) and cleaned beforehand with filtered source water; water and lettuce were changed every day in rearing boxes containing very young snails (≤three weeks of age), or every two days in those with juveniles and adults (≥four weeks of age).

### 2.3. Susceptibility of Snail Egg Embryos to Niclosamide

Egg masses aged 1 to 6 days were collected from each box for each population by cutting out small circles of polystyrene onto which they had attached and examined a stereo zoom microscope SMZ-161 binocular from Motic (SMZ-161-BLED (R2LED) type) to enumerate the number of viable embryos and the extent of embryonation. Only viable eggs containing living embryos were used to assess the susceptibility to Niclosamide (Bayluscide WP 70) (Suzhou Luosen Auxiliaries Co. Ltd., Suzhou, China). Increasing Niclosamide concentrations (0.06, 0.125, 0.25, 0.5 and 1 mg/L) were chosen based on the minimal lethal dose of 0.5 mg/L obtained by Adenusi and Odaibo [17] on *Biomphalaria* eggs species. Egg masses (10 per Niclosamide concentration) were immersed during 24 h in petri dishes, each containing 20 mL of Niclosamide at one of the five concentrations (0.06, 0.125, 0.25, 0.5 and 1 mg/L); egg masses immersed in source water were used as controls. Three replicates of this experiment were used to test any potential variability in the susceptibility to Niclosamide. Overall, a total of 4184 egg embryos were exposed to Niclosamide, and a total of 3849 egg embryos were used as control.

After 24 h exposure, egg masses were removed from Niclosamide solutions and thoroughly washed with source water as recommended by [12] and transferred in 100 mL plastic boxes containing source water. Individual embryos in each egg mass were examined daily for development and hatching during the three following weeks. An embryo was considered dead if its cells became opaque, dull or desegregated [18] or if unhatched at the end of the three weeks experiment.

### 2.4. Susceptibility of Adult Snails to Niclosamide

Two to four egg capsules were collected from each of the 15 isolated boxes for each population and incubated separately in 100 mL boxes. After hatching, juveniles of the first generation (G1), fed ad libitum with lettuces and cleaned beforehand with filtered source water, were reared for 6–8 weeks. G1 adults that have started laying eggs were randomly chosen from each box, and a sample of at least 250 G1 snails was constituted for each population. Groups of 20 snails were randomly immersed in 1 L beakers containing either source water (control groups) or different Niclosamide concentrations (0.06, 0.08, 0.10, 0.12, 0.14, 0.16, 0.18, 0.20 mg/L) (experimental groups); these concentrations were chosen based on the median lethal dose (LC50) of 0.076 mg/L for a *Biomphalaria pfeifferi* population from Nigeria [19]. The beakers were covered with a plastic lid and snails’ avoidance behaviours were observed in the space left between Niclosamide solutions and the lid. After 24 h exposure, snails found at the bottom of the beakers retracted in their shells with or without excretion of reddish haemolymph, which could not move further were considered as dead. Snails still moving were rinsed and placed individually in 100 mL glass vials containing source water and fresh lettuce and checked for mortality for the following two days, since Niclosamide is believed to be degraded within two days post exposure [12]. Since mortality was assessed based on movement of the very tiny snails (about 5 mm), hand lens was used to better appreciate the movements of the snails.

### 2.5. Data Analyses

Statistical analyses were performed using the software Graph Pad Prism6 (GraphPad Software Inc., San Diego, CA, USA). All the variables investigated were categorical (hatching rate, mortality rate, etc.) and were expressed as percentages with 95% confidence interval (CI). Chi-square test was used to compare egg embryo hatching and mortality rates at different Niclosamide concentrations between snail species and populations. Niclosamide toxicity was expressed as LC50 and LC95, corresponding to concentrations that killed 50% and 95% of the exposed snails, respectively. The LC50 and LC95 values with their 95% confidence intervals were determined using probit analysis (Finney 1971) of the mortality data from the susceptibility assay. The threshold for significance was set at 5% for all analyses.

## 3. Results

### 3.1. Susceptibility of Egg Embryos to Niclosamide

The susceptibility to Niclosamide was assessed for a total of 7562 egg embryos originating from 6 snail samples (Table 1). Significant differences were found in hatching rates between samples (Chi-Square = 34.22; *df* = 5; *p* < 0.0001), lower rates being registered for *Biomphalaria pfeifferi* collected in the Mock River and *Bi. camerunensis* collected in Lake Mangoun. For all populations, the hatching rates progressively decreased with Niclosamide concentrations (Chi-Square = 4220; *df* = 5; *p* < 0.0001), no hatching being observed from 0.25 mg/L to 1 mg/L (Table 1). Similarly, significant differences were found in the hatching rates between species (Chi-Square = 1392; *df* = 2; *p* = 0.0009). Such variability was also observed between populations of the same species, either for *Bi. camerunensis* populations (Chi-Square = 9.584; *df* = 1; *p* = 0.002) or for *B. truncatus* populations (Chi-Square = 11.87; *df* = 2; *p* = 0.003).

The LC50 and LC95 were, respectively, 0.171 mg/L and 1.102 mg/L for egg embryos (Table 2); *Bi. pfeifferi* population from Mock River was the most susceptible (LC50: 0.1 mg/L and LC95: 6.3 mg/L), whereas *B. truncatus* population from the Monoun Njindoum Lake was the least susceptible (LC50: 0.20 mg/L and LC95: 1102.53 mg/L).

### 3.2. Susceptibility of Adult Snails to Niclosamide and Avoidance Behaviour

A total of 1130 adult snails aged ~6 weeks were assessed for susceptibility to Niclosamide at a variety of concentrations (0.06, 0.08, 0.10, 0.12, 0.14, 0.16, 0.18, 0.20 mg/L). The mortality rates registered for all *B. truncatus* populations and for all the tested concentrations were 100.0%, significantly higher than the mortality rates observed in *Bi. pfeifferi* and *Bi. camerunensis* populations (Chi-Square = 64.11; df = 2; *p* < 0.0001). In these later two populations, a low mortality rate at 0.06 mg/L (15.0% and 6.0% respectively) and a higher mortality rate at 0.20 mg/L (75.0% and 25.0%, respectively) were registered (Figure 1). Mortality rates increase in general along with concentrations, the peak (87.0%) being reached at 0.20 mg/L (Figure 1). For *Bi. pfeifferi* and *Bi. camerunensis*, although mortality rates were globally increasing according to Niclosamide concentrations, an unexpected decrease was observed for concentrations 0.12 to 0.18 mg/L (Figure 1).

Regarding the lethal doses, *Bi. camerunensis* populations displayed significant differences in susceptibility, snail population from Mangoun being the least susceptible (LC50: 0.20 and LC95: 2.57 mg/L) (Table 2). The LC50 for *Bi. camerunensis* (0.14; 95% CI: 0.11–0.18 mg/L) was similar to the LC50 for *Bi. pfeifferi* (0.14; 95% CI: 0.11–0.18 mg/L). Though slightly higher, the LC95 for *Bi. camerunensis* (0.68; 95% CI: 0.52–0.89 mg/L) was similar to that of *Bi. pfeifferi* (0.60; 95% CI: 0.46–0.76 mg/L). Lethal doses for *B. truncatus* populations were not calculated because the mortality rates were 100% for all the tested Niclosamide concentrations.

### 3.3. Avoidance Behaviour of Adult Snails Exposed to Niclosamide

Adult snails exposed to Niclosamide, in particular *Bi. pfeifferi* and *Bi. camerunensis* species, tended to escape the Niclosamide solutions. This avoidance behaviour included crawling out (distress syndrome) from the walls of the beakers, aggregation at the water–air interface or surfacing behaviour and partial retraction of their head foot. These behaviours were particularly striking in beakers containing the highest concentrations of Niclosamide (≥0.10 mg/L).

## 4. Discussion

This study aimed to assess the susceptibility of six snail samples to increasing Niclosamide concentrations. Snails in these samples reproduce by laying eggs which hatch between 7–10 days [16]. These eggs constitute the dispersal stage for the snail population as they are believed to be the most resistant developmental stage of snail [20,21].

The exposition of egg embryos to Niclosamide revealed an acute toxicity, the latter killing more than 70% of all exposed egg embryos from *Biomphalaria pfeifferi, Biomphalaria camerunensis and Bulinus truncatus*, even at sub-lethal doses (˂0.25 mg/L). Moreover, susceptible embryos never developed beyond the stage they were before exposure, Niclosamide being able to thwart mechanisms involved in the developmental and hatching processes. Higher concentrations are needed for other molluscicide formulations, based on plant extracts, to be lethal for snail egg embryos [22]; in some cases, such molluscicide was proven ineffective to arrest the embryo development in eggs [17]. The Niclosamide ovicidal action was concentration dependent for all exposed populations of the three species. Some eggs exposed to the lowest Niclosamide concentrations hatched, while all those exposed to the highest concentrations (≥0.25 mg/L) did not hatch and contained dead embryos. This observation was already reported in previous studies [23,24] and reveals that a certain threshold concentration of Niclosamide is needed to penetrate the protective coat, made of gelatin-like substance, around egg masses and reach the embryos. This study revealed an important variability in egg embryos susceptibility to Niclosamide within and among species. For example, *Bi. pfeifferi* egg embryos were significatively more susceptible than *Bulinus truncatus* egg embryos, likely because of the genetic differences between these species.

Niclosamide remains the recommended synthetic molluscicide by WHO, as it displays an acute toxicity on all developmental snail stages. In this study, this acute toxicity was observed with adult snails aged approximately 6 weeks, even at low concentrations. A mortality rate of 100% was noticed for *Bulinus truncatus* snails for all tested Niclosamide concentrations. Indeed, it was demonstrated that cell structure defects, inhibition of neurohumoral transmission and energy metabolism caused by the Niclosamide are responsible of the death of another snail species, *Oncomelania hupensis* [25]. The lethality rate in young adults was significantly higher for Niclosamide concentrations ≥0.10 mg/L.

A fluctuation in mortality rates was observed for Niclosamide concentrations between 0.14 and 0.20 mg/L in *Bi. pfeifferi* and *Bi. camerunensis*; this could be explained by the avoidance behaviour exhibited by these exposed *Biomphalaria* individuals. Many of them were able to escape the Niclosamide solution by shelving on beaker walls for many hours. This behaviour, which is a response to loss of water balance [26,27], hinders the action of molluscicide and likely increases the snail’s chance of survival [22,28]. In addition, it was demonstrated that this water-leaving behaviour may be responsible for the recolonization of transmission foci by *Biomphalaria straminea* [13] and *Biomphalaria glabrata* [29,30] after mollusciciding. This avoidance behaviour should be considered for future mollusciciding campaigns against *Biomphalaria* sp. populations in Cameroon.

A significant difference in mortality rates between snail species was observed, *B. truncatus* individuals being significatively more sensitive to lower concentrations of Niclosamide than *Bi. camerunensis* and *Bi. pfeifferi* individuals. The three tested *B. truncatus* populations were collected from different habitat types; two lakes (Petpenoun and Monoun lakes) and one river (Mangoun River). Snail habitat types and human water frequentation rates might not be the only discriminative element explaining this result. Indeed, considering these lower lethal doses, it seems that *Bulinus* sp. snail mollusciciding strategy will be easier to implement since the use of lower doses of Niclosamide could be more approved by populations [30]. To confirm this hypothesis, it will be interesting to screen more *Bulinus* sp. populations, originating from diverse Cameroonian regions.

The susceptibility of Niclosamide was quite variable between populations, with *Bi. camerunensis* snails collected in the Lake Petpenoun and Mangoun River being the more resistant populations. Previous molecular studies on *Bi. camerunensis* populations originating from these sampling sites revealed a high genetic diversity compared to other *Bi. camerunensis* and *Bi. pfeifferi* populations [14]. This might explain the variable level of susceptibility of snail hosts to Niclosamide, as it was demonstrated that it can be due to differences in natural tolerance among geographic isolates rather than to selection of resistant strains. This is supported by the fact that a high resistance to a strain of *Schistosoma mansoni* was also observed in Petpenoun population [31]. Repeated mollusciciding in such populations, more than twice annual applications could be necessary to reduce snail population density below a critical threshold to sustain transmission [19].

The toxicity of Niclosamide was different between developmental stages of snail species and populations, adults being significantly more susceptible than eggs. In *B. truncatus* species, for example, 100% of adults were killed by a dose of 0.06 mg/L while their egg embryos were only susceptible at concentrations equal and above 0.25 mg/L, the egg protective coat being efficient to protect embryos at lower concentrations. This finding suggests that Niclosamide concentrations to be used for mollusciciding should be estimated from testing on egg embryos, which appeared less susceptible than adults in this study. This decision should, however, take into account both the possible side effects of Niclosamide on aquatic fauna (fishes, frogs and tadpoles) [12,31] and the poor community acceptability when Niclosamide is applied at higher environmental doses, such as 1 mg/L [30].

Alternatives to molluscicides, such as biological controls, including the use of competitors or predators, have also been proven efficient and less (or not) toxic compared to Niclosamide. However, the large-scale implementation of these approaches is yet to be developed, and, until now, mollusciciding/chemical control is still the more efficient strategy for snail control [12]. Previous studies suggested that repeated mollusciciding, if properly done, even with lower doses (0.25 g/m^3^), could be efficient to kill schistosome larvae in water [32] and reduce snail populations [9,33]. This dose (0.25 g/m^3^) is the optimal one found in this study for egg embryos and could be used in combination with other control strategies recommended by WHO [34,35,36,37], to accelerate the elimination of schistosomiasis.

## 5. Limitations

The main limitation of this study is the fact that the identification of snails was performed using morphological features, which is sufficient to classify species, but might not be enough to clearly distinguish strains, unlike molecular approaches. This is important because the susceptibility of snails to Niclosamide can be associated with the strains of the latter. Indeed, molecular identification has been carried out on some snails collected in the some of the sites where our study has been conducted, and the results revealed distinguished haplotypes among snail species [14].

The second limitation of this study is the fact that the death of snails was appreciated based only on the movement of the latter, even if hand lens were used for more accuracy. Histological studies and electron microscopy would have been better to reinforce and validate our observations. However, snails suspected to be dead were put aside and observed for two consecutive days to ascertain that they were indeed dead, thus mitigating the logistical constrains for such confirmatory analyses.

## 6. Conclusions

This study revealed an important variability in the susceptibility to Niclosamide for the six snail samples tested. Adult snails appeared to be more susceptible than eggs, and specific concentrations at which Niclosamide is effective against both egg and adult snail stages were identified. The ovicidal dose seems then to be the better dose for mollusciciding of these snail populations. The high variability in susceptibility to Niclosamide displayed by the different species, populations and developmental stages of snails suggests that mollusciciding should not be done without appropriate pre-control studies.

## Figures and Tables

**Figure 1 ijerph-19-12873-f001:**
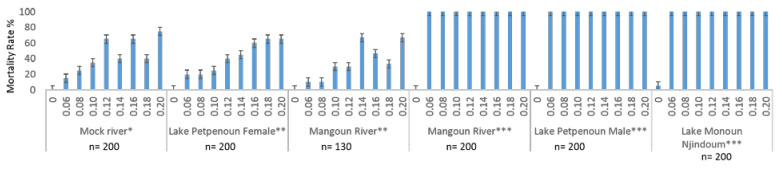
Mortality rates of adults to different Niclosamide (Bayluscide WP 70) doses according to snail species. Figures on the *x* axis represent the different concentrations of Niclosamide, “0” being used for controls.*: *Biomphalaria pfeifferi*; **: *Biomphalaria camerunensis*; ***: *Bulinus truncatus*; n: number of tested snails.

**Table 1 ijerph-19-12873-t001:** Effect of different Niclosamide (Bayluscide WP 70) concentrations on snail egg embryos according to sampling sites and species.

Snail Sampling Sites	Niclosamide Dose (mg/L)					
	**0**	**0.06**	**0.125**	**0.25**	**0.5**	**1**
***Mock River*** *						
No. egg masses	55	12	12	12	12	
No. egg embryos	483	106	129	134	116	12
% egg embryos hatched (95% CI)	90.5 (87.5–92.8)	66.6 (56.6–74.4)	0 (0.0–3.5)	0 (0.0–2.8)	0 (0.0–3.2)	134
***Lake Petpenoun Female*** **						
No. egg masses	34	10	11	10	11	0 (0–2.8)
No. egg embryos	766	210	233	248	277	11
% egg embryos hatched (95% CI)	99.0 (98.0–99.5)	91.9 (87.4–94.9)	86.3 (81.3–90.1)	0.0 (0.0–1.5)	0.0 (0.0–1.4)	232
***Mangoun River*** *						
No. egg masses	37	4	4	4	5	0.0 (0.0–1.6)
No. egg embryos	478	83	59	85	68	5
% egg embryos hatched (95% CI)	96.7 (94.6–97.9)	65.1 (54.3–74.4)	3.4 (0.9–11.5)	0.0 (0.0–4.3)	0.0 (0.0–5.3)	111
***Mangoun River*** ***						
No. egg masses	42	15	15	16	16	0.0 (0.0–3.3)
No. egg embryos	621	141	159	186	198	16
% egg embryos hatched (95% CI)	97.9 (96.5–98.8)	83.0 (75.9–88.3)	66.7 (59.0–73.5)	0.0 (0.0–2.0)	0.0 (0.0–1.9)	218
***Lake Petpenoun Male*** ***						
No. egg masses	40	9	12	10	10	0.0 (0.0–1.7)
No. egg embryos	422	119	146	121	141	10
% egg embryos hatched (95% CI)	89.3 (86.0–91.9)	54.6 (45.7–63.3)	89.7 (83.7–93.7)	0.0 (0.0–3.1)	0.0 (0.0–2.7)	153
***Lake Monoun Njindoum*** ***						
No. egg masses	48	9	8	8	9	0.0 (0.0–2.4)
No. egg embryos	483	124	109	86	94	8
% egg embryos hatched (95% CI)	93.4 (90.8–95.3)	75.0 (66.7–81.8)	63.3 (53.9–71.8)	0.0 (0.0–4.3)	0.0 (0.0–3.9)	89

No.: Number of; *: *Biomphalaria pfeifferi*; **: *Biomphalaria camerunensis*; ***: *Bulinus truncatus*; CI: confidence interval.

**Table 2 ijerph-19-12873-t002:** Lethal concentrations (**LC50** and **LC95**) of egg embryos and adult snails to Niclosamide (Bayluscide WP 70) according to sampling sites and species.

Snail Sampling Sites	LC50 (95% CI)	LC95 (95% CI)
** *Egg embryos* **		
Mock River *	0.10 (0.02–0.20)	6.30 (2.50–15.60)
Lake Petpenoun Female **	0.84 (0.47–1.49)	8.07 (4.52–14.39)
Mangoun River **	0.11 (0.05–0.22)	2.64 (1.28–5.5)
Lake Petpenoun Male ***	0.27 (0.18–0.42)	2.10 (1.35–3.25)
Mangoun River ***	0.81 (0.38–1.72)	20.72 (9.76–44.01)
Lake Monoun Njindoum ***	0.20 (0.03–1.29)	1102.53 (174.27–6975.34)
** *Adult snails* **		
River Mock *	0.14 (0.11–0.18)	0.60 (0.46–0.76)
Lake Petpenoun Female **	0.14 (0.11–1,18)	0.56 (0.44–0.71)
Mangoun River **	0.20 (0.13–0.30)	2.57 (1.68–3.94)

LC50: lethal concentration 50; LC95: lethal concentration 95; *: *Biomphalaria pfeifferi*; **: *Biomphalaria camerunensis*; ***: *Bulinus truncatus*.

## Data Availability

The datasets used and/or analysed during the current study are available from the corresponding author on reasonable request.

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
