# Peer review of "Exposition of Intermediate Hosts of Schistosomes to Niclosamide (Bayluscide WP 70) Revealed Significant Variations in Mortality Rates: Implications for Vector Control"

_ijerph, 2022, doi:10.3390/ijerph191912873_

Round 1

Reviewer 1 Report

Investigate vector control strategies is very interesting and necessary. However, important questions must be answered. 

Introduction - Pag 2, line 67.

Elimination of intermediate snail hosts from local habitats may be efficient for the prevention of Schistosoma infection and/or reinfection.

-Please, replace "elimination" for control.

 Materials and Methods

Susceptibility of snail egg embryos to Niclosamide – page 03

 “Egg masses (10 per Niclosamide concentration) were immersed during 24 hours in petri dishes, each containing 20 mL of Niclosamide at one of the five concentrations (0.06, 0.125, 0.25, 0.5 and 1 mg/L)” – line 121- 122.

 Question: How would the applicability in the environment be? Is this experiment viable?

 Susceptibility of adult snails to Niclosamide

 "Groups of 20 snails were randomly immersed in 1L beakers containing either source water (control groups) or different Niclosamide concentrations (0.06, 0.08, 0.10, 0.12, 0.14, 0.16, 0.18, 0.20 mg/L)". line 137- 139.

Questions: How would the applicability in the environment be? Is this experiment viable?

               How toxic would niclosamide be to aquatic flora?

              How toxic would niclosamide be to marine environment?

Results:

3.1. Susceptibility of egg embryos to Niclosamide

Susceptibility of adult snails to Niclosamide and avoidance behaviour- Page 05

 Question : Histological studies and Electron Microscopy can reinforce and validate the data.

Reviewer 2 Report

Overall, the article titled, "Exposition of intermediate host of schistosomes to Balusicide WP 70 revealed significant variation in mortality rates; implications for vector control was written well. Include, the manuscript lacks overall organization (e.g. structured paragraphs and handling of taxonomic genera), details in methods, and was extremely lights on citations (in some cases could consider plagiarism if not cited). I do feel this work is important and if these major and minor issues are fixed, I believe the manuscript could be a strong piece of literature.

General edits:

·      All taxonomic names must be in italics

·      Genera can not stand alone—either include a species or write sp. or spp. or say genera X

·      Each paragraph should a topic sentence

·      Pick one way to refer to the molluscicide and use that throughout the entire manuscript – either Niclosamide or Bayluscide WP 70—as it currently written very confusing to the reader

·      Cant just say a procedure was done at random—how was it random?

·      All table and figure legends should be able to stand alone. All details are not included in any table or figure in this manuscript.

Line by line or section edits:

Title needs to be re-organized or recast. The semi colon doesn’t work in this case. Also doesn’t really fit the body work well.

Introduction:

Paragraph one is missing a lot of citations. Each uncited sentence in this paragraph is not original to the authors of this manuscript and citations need to be included (sentence 1-3). These errors are borderline acts of plagiarism.

Line 71 needs a citation

Line 71-73 Physa needs italics and cant stand alone—see above for explanation

Line 77 -79 is stating results, I think or it is poorly written. Recast so it is clear and is not providing results.

Line 78 Need the manufacture for Bayluscide WP 70

Line 81 missing a citation

Line 83-86 is supplying the aims or question this study is trying to examine. It is not clear and should be worded such that the reader understands what are the aim of the study— when I read this study (looking at the results) there are three aims that should be clearly stated here. Such as, “Here we aim to experimentally test 1)jakdhkfjkajkjkl, 2)hfhadhakjk, and 3)dherhoehla.

Predications to these aims can be provided following listing them or not. But the paragraph should include a sentence or two about boarder impacts of the findings. Why should the reader care about this study?

 Methods:

Overall these methods need the most improving. At there current state they could not be repeated nor is clear to the reader how the study was performed.

Add a section called “Study system”

          Within this section provide the reader details necessary to understanding the snails and cercariae. Support with citations.

Lines 89-91 are fine but the next sentences should explain why these sites were chosen. Explain in detail—as you compare the sites in results so they need to be supported in methods. How many person hours were used to collect the snails, 

Lines 92-94 these are methods not developed by authors and should be cited

Line 94 Details should be provided on how snails were identified- list morphology keys used as well as provide key characteristics. Also for each snails species a museum voucher should be submitted. Currently morphology should not stand alone to determine taxonomic identification of a snail—molecular data should be performed, and those details should be included in the next revision. Especially because in discussion there is comments about cryptic species. If molecular work is not performed it must be very clear to the reader that morphology supported taxonomic identification. Details and citations must be included.

Line 101-102 What snails were maintained on should be discussed.

I am bit confused here—wild caught adults snails are likely to be infected with multiple different trematodes. Why would adults be placed together without infections beginning checked. This is huge issue, if I understand that these were the same adult snails used in the experiments as infected snails could respond differently to the Niclosamide than non-infected snails. Were any snails cracked as a subset to make sure no infections occurred? If so, these should be listed in the methods.

Line 103 its unclear what a plastic box is—maybe include photo or explain if the top was on each box. Also, some plastic leaches into water – was this food grade? What water was used? Was the water treated or was chlorine removed? If source water was used—was it filtered for cercariae and other free living things?

Line `104- here is some discussion on cercariae but no details—how was performed? Cite and include the methods. Were snails that tested positive removed from study? This is deferential to the results—without this piece the results are iffy and unsupported as the seen variation could have been from infection or noninfected verse infected.

Line 104-105 Describe what is meant by randomly chosen. Also describe what is means but mature adults—how can this known from a snail?

Line 110 was lettuce cleaned?

Line 116 manufacture and type details are necessary for scope used

Lines 120-125 the number of snails should be reported here. Its hard to follow how many snail or egg masses we are taking about. After reporting each treatment was replicated three time list the total number of snails.

Line 127 what does it mean by washed in source water—explain this procedure and why this is necessary part of the methods—cite if this is common practice in other toxicology studies

Line 129 why 3 weeks – explain

Line 135 this time it says clean lettuce—what does this mean and cleaned with what? Organic?

Line 135 were water changes performed in this part of the rearing – how often?

Line 136 and 138 explain what is meant by randomly chosen—be specific – how

Line 137 how was 20 picked—was there a power analysis on for number of individuals completed? If not this should be done to explain why 20 is an alright number of snails

Line `144  snails do not have blood. Snails have hemolymph

Line 147—explain why two days  only checked for mortality—what happens after? How long were the snails maintained?

Explain the procedure for checking mortality with a hand lens—this is unclear. What are you looking for? What deems a snail dead with this method? Cite this method.

Lines 149- 158 should include details on how the power analysis was done to determine sample group size

Line 176—Legend needs to include concentrations, site of snail eggs embroys and make sure be consistent with naming of Niclosamide. Remove lines from table. Not the sample sites vary in how the name is written between figure 1 and both tables—either capitalize Female and Male or don’t. Be consistent. In the explanation of the ** Scientific names are lacking italics

Line 180 I don’t understand why they are called young adults now. What does this mean?

Line 197-199 Why!

Figure 1—caption needs stand alone. Sample numbers, doses and life stage of snails should be included. These populations do not match the text or Table 1. Keep the naming consistent. It is hard to see delineation of between species—which should say taxa since you are working with two genera. These taxonomic names should not be abbreviated. Should include SD or SE bars and these should be details in the methods. Remove horizonal lines.

Table 2 – caption needs to stand alone. Why are using 2 significant digits here when in Table 1 you used 1. It be helpful to provide error estimates and sample sizes here as well. End dash is preferred to hyphen. Scientific names are not in italics in subnote.

Lines 206-212 Were not detailed in the AIMS. Add to the aims. Line 208-212 is not  result its an explanation should be left in discussion.

Lines 213-219 should be removed and replaced with the novel findings of this experiment. Make sure topic sentence is supported.

Line 222 should be taxa as its multiple genera

Line 233-234 is actually the topic sentence to this paragraph.

Line 245 why young adults now—you have bounced back and forth with the terms adult and young adult

Lines 238-246 needs a topic sentence to the paragraph

Line 256 genus cannot stand alone

Line 255- remove “ be taken into account”

Line 263—what are the other options? Cite literature and compare and contrast to your findings.

Line 270-274 makes it very clear that molecular identification of snails must occur for this study to be able to be applied without it lack of application for these results

Line 284 confused! Adults have cercariae

Lines 288-232 cant support a paragraph alone. Expand the paragraph by providing a example or two found in the literature

Reviewer 3 Report

The manuscript is well written and the topic is of interest. The research on anthelmintics to confront schistosomiasis and the possible resistances should be further explored. There are other similar papers, but the present one apports new insigths on this topic. 

Minor revisions:

Line 135: 6-8 weeks

Line 178 and 204-205: Lack of italics

Line 200: Figure caption shoul be under the figure 

Round 2

Reviewer 2 Report

I thank the authors for their clear, concise and detailed review letter. I believe they considered my concerns as well as the other two reviewers well. For which now I believe the manuscript is sound and in form to be submitted. With that said a detailed grammar check should still be considered by the authors for minor errors.